# Evaluation of Fermented Oat and Black Soldier Fly Larva as Food Ingredients in Senior Dog Diets

**DOI:** 10.3390/ani11123509

**Published:** 2021-12-09

**Authors:** Kangmin Seo, Hyun-Woo Cho, Julan Chun, Junghwan Jeon, Chanho Kim, Minji Kim, Kwanho Park, Kihyun Kim

**Affiliations:** 1Animal Welfare Research Team, National Institute of Animal Science, Rural Development Administration, Wanju 55365, Korea; kmseo@korea.kr (K.S.); jhwoo3856@korea.kr (H.-W.C.); julanchun@korea.kr (J.C.); jeon75@korea.kr (J.J.); kch8059@korea.kr (C.K.); 2Division of Animal Nutrition and Physiology, National Institute of Animal Science, Rural Development Administration, Wanju 55365, Korea; mjkim00@korea.kr; 3Industrial Insect Division, National Institute of Agricultural Sciences, Rural Development Administration, Wanju 55365, Korea; nicegano@korea.kr

**Keywords:** canine, insect, fermented product, alternative ingredient, pet food

## Abstract

**Simple Summary:**

Along with concerns about the shortage of future food resources, the problem of ensuring a stable supply of feed materials is emerging. The rapid growth of the pet food market is also increasing the demand for new food ingredients, requiring the evaluation of their safety and nutritional value. Recently, insects and fermented foods are some of the materials that have entered the spotlight as potential future foods, and studies on their usefulness as food are being actively conducted. This study aimed to evaluate and verify the safety of fermented oat (*Avena sativa*) and black soldier fly larva (*Hermetia illucens* L.) when used in a dog food as part of the effort toward discovering suitable nutritionally excellent and functional food materials. Our results show that 10% fermented oat flour, 5% black soldier fly larva meal, or a combination thereof in the food did not negatively affect food intake, body weight, fecal status, skin condition, or hematological and biochemical parameters. Overall, our findings suggest that fermented oat and black soldier fly larva can be used as food ingredients for dogs.

**Abstract:**

The aim of this study was to evaluate the suitability of fermented oat (FO) and black soldier fly larva (BSFL) as food ingredients for dogs. A total of 20 spayed female dogs were divided into four treatment groups, with 5 dogs per group. The four treatment groups consisted of a control group, a diet with 10% FO, one with 5% BSFL, and one with 10% FO and 5% BSFL, and each experimental food was fed for 12 weeks. The feeding of FO and/or BSFL did not affect the daily food intake, body weight, body condition score, fecal score, or skin condition of the dogs. In all the experimental groups, no significant differences in serum IgG, IL-10, or TNF-α levels were observed upon the feeding of FO and/or BSFL. Some hematological (white blood cell and basophils) and serum biochemical parameters (phosphorous, globulin, and alkaline phosphatase) showed significant differences with FO and/or BSFL feeding compared to the control group, but they were within the normal reference range. No adverse clinical signs related to these parameters being affected by FO and BSFL were observed. The feeding of BSFL for 12 weeks reduced the serum cholesterol level (*p* < 0.05) at the end of the experiment. Our findings suggest the suitability of FO and BSFL as food materials for dogs.

## 1. Introduction

Amid increasing global concerns over the shortage of future food resources [1,2], the rapid growth of the pet food market is raising concerns regarding whether a stable supply of raw materials for pet food can be maintained. In addition, hypercompetition in the pet food market has led to the indiscriminate use of new ingredients where preliminary verifications of safety and nutritional value are lacking [3]. In this regard, the need for research on the nutritional value, safety, and functionality of novel ingredients that can replace existing ingredients has been emphasized [4].

In the pet food industry, livestock products are mostly used as a protein source, but the demand for novel protein materials is increasing due to the competitively for their use in food for humans, and regarding the sustainability of livestock products. In other words, it is known that livestock products mainly used as protein sources in dog food can cause allergies. Therefore, many efforts, such as towards using hydrolyzed proteins and finding alternative protein ingredients, have been made to reduce the allergic response induced by exposure to protein sources in dogs [5]. Many researchers are paying attention to the potential suitability of edible insects for human food as well as animal feed [6,7,8]. In particular, black soldier fly larva (BSFL; *Hermetia illucens* L.) has been reported to be a suitable insect species given its nutritional value, safety, and amenability for mass production [9,10]. Several studies have reported that BSFL meal can partially replace major protein sources (e.g., fish and soybean meal) in conventional diets for poultry [11,12], fish [13,14], and pigs [15]. In addition, some recent studies that evaluated the safety and physiological effects of a BSFL diet on companion dogs have reported positive results [16,17,18]. Nonetheless, more research is needed in terms of the safety and feeding effects of using BSFL in pet food.

Recently, the trend in the pet food market is showing a shift toward grain-free, gluten-free, human-grade, natural, and organic pet food. However, scientific evidence that these are nutritionally superior or that they are more beneficial to pet’s health is lacking. Oats, one of the grains, are known to provide nutrients such as proteins, unsaturated fatty acids, vitamins and minerals, as well as arabinoxylan, β-glucan, and phenolic compounds, having multiple functional and bioactive properties [19,20]. Although there are very limited studies on the efficacy of oat feeding in dogs, one study reported that the intake of oat beta-glucan could improve the apparent total tract digestibility of macronutrients, and was effective in reducing serum total cholesterol and low-density lipoproteins in adult dogs [21]. Meanwhile, although not studied in dogs, previous studies have suggested that oats with various biologically active substances help to prevent diseases, such as cardiovascular pathologies, colon cancer, type II diabetes, and obesity in humans [22,23,24,25,26]. Furthermore, attempts have been devoted to developing oat-based fermented foods using lactic acid bacteria to improve the nutritional value and functionality of oats [27,28,29]; some studies have confirmed the potential value of fermented oats as a functional food [30,31,32]. Despite the positive effects of oats or fermented oats (FOs), no study has reported the effects of feeding FOs to dogs. Therefore, this study was conducted to evaluate the safety and feeding effects of FO and BSFL in dogs.

## 2. Materials and Methods

### 2.1. BSFL and FO Preparation

Freeze-dried BSFLs were supplied from the National Institute of Crop Science (Wanju, Republic of Korea). After hatching, the 5-day-old BSFLs were bred with corn and soybean meal-based feed (19% crude protein and 3150 ME kcal/kg) until 17 days of age, and produced by washing and freeze-drying. FO was prepared as follows: the whole-grain oats (*Avena sativa*) were ground with water, and incubated for 8 h at 37 °C with a starter (1 × 10^8^ CFU/mL *Pediococcus pentosaceus*, CBT SL4; 5 × 10^8^ CFU/mL, *Bifidobacterium longum*, KCTC 10630BP; 5 × 10^8^ CFU/mL, *Lactobacillus plantarum*, KCTC 1048). Thereafter, the supernatant was removed by centrifugation, and the precipitate was dried and used for the experimental diet.

### 2.2. Animals, Designs, Diets, and Housing

This experiment was conducted in accordance with the method approved by the Animal Care and Use Committee National Institute of Animal Science (NIAS-2018-308). Twenty spayed, 10.8 ± 0.04-year-old, small-breed dogs (seven Schnauzers, six Poodles, and seven Maltese; initial body weight (BW) 4.18 ± 0.32 kg; 9-scale body condition score (BCS) 4.2 ± 0.17) were used in this study. The dogs were randomly divided into four groups with 5 dogs per group: Group 1—fed a rice and poultry meal-based diet (CON); Group 2—fed a diet with 10% FO (FO); Group 3—fed a diet with 5% BSFL (BSFL); Group 4—fed a diet with 10% FO and 5% BSFL (FO+BSFL). In this study, there were many variables, including three breeds of dogs and four treatment foods, but the number of dogs assigned to each group was as small as five. Thus, only female dogs were used to minimize other variables. All experimental diets were formulated to meet the nutritional requirements for adult dogs as suggested by the Association of American Feed Control Officials [33] (Table 1). The inclusion rates of FO and BSFL were determined to be within the level that could meet both isonitrogenous and isocaloric criteria as the control food. All ingredients for the experimental food were used based on commercial products in a powder type, except for lard. All ingredients were mixed at 500 rpm for 10 min using a food paste mixer (Mixer, Sung-il, Seoul, Korea), and then liquid lard and water were added and kneaded at 1500 rpm for 20 min. The mixed dough was subdivided into pieces of 10 cm in diameter and steam-heated (100 °C or above) for 40 min using a steamer (Rice cake maker, Dahan, Seoul, Korea). Thereafter, the steamed dough was pelleted into a cylindrical shape with a diameter of 10 mm and a length of 15 mm using a noodle machine (Noodle maker, Yusung, Daegu, Korea). The pelleted food was dried in a dryer at 70 °C for 1 h and stored at −20 °C until feeding (dry-oven, Shiniltech, Jeonju, Korea). The experimental food was transferred from the freezer to room temperature 1 day before feeding and allowed to cool until it reached room temperature before feeding. No palatants were used in this experimental food.

The chemical compositional parameters of the experimental foods were analyzed following standard Association of Official Analytical Chemist (AOAC, 2006) methods [34], including the moisture content (AOAC method 934.01), crude protein (CP, AOAC method 984.13), ether extract (EE, AOAC method 920.39), ash (AOAC method 942.05), crude fiber (CF, AOAC method 978.10), calcium (Ca, AOAC method 927.02), and phosphorus (P, AOAC method 965.17). The nitrogen-free extract (NFE) was calculated using the following equation: NFE (% DM) = 100 − (CP + CF + EE + Ash).

Each dog was housed in an individual room (1.7 m × 2.1 m) at consistent room temperature (22–24 °C) and with consistent lighting (12 h light and 12 h dark cycle) for the study period. Food was provided at an amount estimated by the MER equation for each dog twice per day throughout the duration of the trial, and drinking water was provided ad libitum for 12 weeks. The MERs of the dogs were calculated using the AAFCO’s MER calculation as follows: MER = 132 × metabolic body weight (mBW). Food intake was measured daily, and BW and BCS were measured weekly. The rate of change in body weight gain (BWG) was calculated as follows: rate of change of BWG = (final body weight)/(initial body weight) × 100. BCS was evaluated on a 9-point body condition score scale according to the criteria developed by Laflamme [35]. Fecal scores were evaluated on a 5-point fecal score scale (1 = dry to 5 = liquid feces) according to the Waltham Fecal Scoring System [36] every day during the entire test period.

### 2.3. Sampling and Analysis

Blood samples were collected from the jugular vein after 12 h of fasting at the beginning and end of the experiment. The collected blood was immediately separated into EDTA vacutainer tubes (ref 367861, BD Vacutainer, NJ, USA) and serum vacutainer tubes (ref 367812, BD Vacutainer, NJ, USA). The whole blood in the EDTA vacutainer tubes was used for complete blood cell count (CBC) analysis immediately after collection. CBCs were measured using an automatic hematology analyzer (IDEXX Laboratories, Inc., Westbrook, ME, USA). Serum was obtained by centrifugation (2000× *g*, 10 min) from blood in the serum vacutainer tubes, and then stored frozen (−80 °C) until analysis. The serum biochemical parameters were analyzed using an automatic biochemical analyzer (Hitachi 7180; Hitachi High-Technologies Co., Tokyo, Japan). Serum canine tumor necrosis factor-alpha (TNF-α, SEKC-0033, Solarbio Co., Ltd., Beijing, China), interleukin-10 (IL-10, SEKC-0026, Solarbio Co., Ltd., Beijing, China), and immunoglobulin G (IgG, SEKC-0050, Solarbio Co., Ltd., Beijing, China) were quantified using enzyme-linked immunosorbent assay kits according to the manufacturer’s instructions.

The transepidermal water loss (TEWL), moisture, and oil content in skin were measured to examine the skin conditions in the groin, armpits, and ears at the end of the experiment (at 12 weeks) using a closed chamber-type instrument (GPSkin Barrier^®^, GPOWER Inc., Seoul, Korea). The hair in each skin area was removed with a dog hair clipper without using a cleanser 1 day before the measurement.

### 2.4. Statistical Analysis

All statistical analysis was performed using SPSS version 17.0 (SPSS Statistics, IL, USA, 2009). Data are presented as mean ± standard error (SE). Because all experimental groups were composed of three breeds, the significant differences among the control group (CON) and each test group (FO, BSFL, and BSFL + FO) were analyzed by univariate analysis with a general linear model (GLM) with Tukey test as the post-hoc analysis, treating the breed factor as a covariate. Changes in CBCs and serum biochemical parameters over time were analyzed using repeated measures of GLM. The associations between the experimental foods and BCS or fecal score were investigated using the nonparametric test with a Chi-squared test. Differences were considered statistically significant when *p* < 0.05.

## 3. Results

### 3.1. Food Intake, Body Parameters, and Fecal Score

Table 2 shows the daily food intake, BW, and BCS, for which no significant differences were found among the CON and each treatment group. In all groups, the body weight was higher at the end of the experiment than at the beginning, but there was no significant difference among the control dogs and those fed FO and BSFL.

The fecal scores of all the experimental groups were within the desirable range of between 2.10 and 2.40, and the dog food with FO and BSFL did not affect the fecal scores (Figure 1a,b).

### 3.2. Skin Status

Figure 2 shows the effects of the feeding of FO and BSFL on skin status. At the end of the experiment, the TEWL, moisture, and oil levels of the groin, armpits, back, and ears were measured, and the measured value is expressed as the average of values determined at the four sites. The TEWL, moisture, and oil levels of each treatment group did not show significant differences compared to the control group (Figure 2a–c).

### 3.3. Hematological and Biochemical Parameters

The results of hematological parameters are presented in Table 3. All hematological parameters analyzed in this study were within the normal reference range, and no significant differences in these parameters were observed by the single or combined feeding of BSFL and FO among all experimental groups, except for white blood cells (WBCs) in the BSFL group. At the end of the experiment, the BSFL group had a significantly higher WBC value than the CON group (*p* < 0.05). Basophils (BASO) were not affected by the feeding FO or BSFL, but BASO in the BSFL group was significantly increased at the end of experiment compared to the beginning (*p* < 0.05). All the experimental groups showed no significant effects of FO and BSFL on neutrophils (NEU), lymphocytes (LYM), monocytes (MONO), red blood cells (RBC), hemoglobin (HGB), or hematocrit (HCT) during the study period.

The results of serum biochemical parameters are presented in Table 4. During the experimental period, no significant effect of feeding FO or BSFL was observed on serum glucose (GLU), creatinine (CREA), blood urea nitrogen (BUN), calcium (CA), alanine aminotransferase (ALT), gamma glutamyltransferase (GGT), or albumin/globulin (A/G) ratio values among all experimental groups. However, the FO and FO + BSFL groups showed significantly lower alkaline phosphatase (ALKP) than the control group at the end of the experiment (*p* < 0.05). For the FO + BSFL group, we recorded significantly lower GLOB compared to the control group at the end of the experiment (*p* < 0.05). For the BSFL group, we recorded significantly lower phosphorous (PHOS) compared to the control group (*p* < 0.05). In addition, the BSFL group showed significantly less total protein (T-PRO) and total cholesterol (T-CHO) at the end compared to the beginning of the experiment (*p* < 0.05).

The values for all the parameters of the FO and FO + BSFL groups remained within the reference ranges throughout the experiment. However, the control group showed slightly higher GLOB values both at the beginning (3.82 ± 0.28 g/dL) and at the end of the experiment (4.12 ± 0.23 g/dL) compared to the reference range (1.6–3.6 g/dL). Although the BSFL group showed slightly higher GLOB (3.80 ± 0.17 g/dL) and T-BIL (0.40 ± 0.03 mg/dL) values compared to the reference ranges (GLOB, 1.6–3.6 g/dL; T-BIL, 0.1–0.3 g/dL) at the beginning of the experiment, they were measured as being within the normal ranges at the end of the experiment (Table 4).

### 3.4. IgG and Cytokines

Figure 3 shows the effects of feeding FO and BSFL on changes in canine immunoglobulin G (IgG), interleukin 10 (IL-10), and tumor necrosis factor alpha (TNF-α) levels. Canine IgG ranged between 1.68 and 8.10 mg/mL, the IL-10 ranged between 140.24 and 415.58 pg/mL, and the TNF-α between 1.20 and 6.12 pg/mL, and no statistically significant differences were found in any of the experimental groups compared to the control group. In addition, no significant changes were observed in the levels of IgG, IL-10, or TNF-α in any experimental group for 12 weeks (initial vs. final; Figure 3).

## 4. Discussion

### 4.1. Feeding and Body Parameters

This study was performed to evaluate the suitability of BSFL and FO for inclusion in a dog diet. Many studies have reported the effects of feeding BSFL [10,11,14] to livestock and dogs; however, to the best of our knowledge, this study is the first to evaluate FO as a food ingredient for dogs. We confirmed that dietary supplementation of 5% BSFL, 10% FO, and 5% BSFL + 10% FO had no effect on food intake, body weight, or BCS in dogs. These results are consistent with those of Freel et al., who reported that the feeding of defatted BSFL meal (5%, 10%, or 20%) and BSFL oil (2.5% or 5%) for 4 weeks did not affect food intake or BW in adult beagle dogs [18]. In addition, although the focus was not fermented oat, Traughber et al. reported that feeding a diet containing 40% oats did not affect food intake or BW compared with the dogs fed a diet containing 40% rice [37]. Moreover, in dogs fed a diet containing 25% barley, which has a composition similar to oats, there was no effect on daily intake, preference ratio, nutrient digestibility, or stool scores compared to those with a diet containing 25% rice [38]. When the basic diet was a vegetarian diet, supplementation with corn (20.1%), rye (20.1%), and fermented rye (60.4%) was found to affect food intake and fecal scores, but did not affect body weight in dogs [39]. Fermented food is known to be less palatable for dogs due to its unique smell and taste [40], but we did not find a significant effect of dietary supplementation with FO on food intake. In the study by Lee et al., chicken meat fermented with *Pediococcus* spp. had a lower palatability than non-fermented chicken meat, but, nevertheless, there were no changes in diet intake or body weight in their study, and their results were consistent with our findings [41]. FO was obtained by centrifugation in this study. We inferred that the reason why FO feeding had no effect on food intake could be because some of the offensive compounds affecting palatability might be removed during the supernatant removal process after centrifugation. Although we did not analyze the nutrient digestibility of the food used in this study, the lack of change in BW and BCS in dogs fed the experimental diets for 12 weeks, together with the fact that MER was met, suggests that BSFL and FO have value for use as food materials that do not have a negative effect on nutrient availability.

### 4.2. Safety and Health Parameters

A food allergy is defined as a hypersensitivity response caused by an abnormal immune system response to a specific allergen in the food; about 1% of canines and felines have an allergic reaction to food [5]. This food allergy in dogs is mainly caused by protein sources derived from livestock products (such as chicken, beef, lamb, and egg etc.) [5]. The main clinical signs of food allergy and hypersensitivity are dermatological disease (e.g., pruritus, erythema, papular eruptions, etc.) and inflammatory response, accompanied by gastrointestinal symptoms (e.g., vomiting, diarrhea, frequent defecation, colitis, etc.) [5,42]. To evaluate the clinical signs of allergy to dietary BSFL and FO, we investigated the skin status (TEWL and contents of moisture and oil), fecal score, and immune-related parameters (IgG, IL-10, and TNF-α) in serum. TEWL is the amount of water lost through the skin epidermal layer. An increase in TEWL indicates an impairment of the skin barrier function [43], and Shimada et al. suggested that the TEWL can be used as an indicator reflecting the damaged functioning of the skin barrier in dogs [44]. In addition, the pathophysiological response of food allergy and/or hypersensitivity is caused by immune action due to the allergens that have passed through the intestinal mucosal barrier in gut-associated lymphoid tissue. The immune response is induced by the interaction between immunocytes (e.g., Mast cells, eosinophils, helper T cell 1, helper T cell 2, etc.) and immunoglobulins (e.g., IgE, IgA, IgG, etc.) and various cytokines (e.g., IL-4, IL-5, IL-10, IL-6, TGFβ-1, TNF-α, etc.) [45,46]. The results of this study show that there was no significant effect on the skin status, fecal score, or immune parameters (IgG, IL-10, and TNF-α) of dogs fed a diet containing 5% BSFL and 10% FO for 12 weeks. These results suggest that BSFL and FO, as food ingredients, pose a low risk as allergens to dogs. The process of fermentation is a representative process for beneficial health food, and is known to enhance detoxification and suppress allergic reactions by reducing aflatoxins and producing antimicrobial factors [47]. Park et al. [48] reported that the addition of medicinal plants fermented by *Enterococcus faecium* to dog food results in antioxidant activities and improves the fecal microbiota, with a higher number of beneficial microorganisms in dogs. One study demonstrated that fermented soybean products enhance the activity of natural killer cells and increase TNF-α gene expression in antigen-stimulated PBMCs in dogs [49]. Although it was not performed using dogs, a previous study showed that the intake of fermented wheat bran in pigs increased the protein expression of IL-10 [50]. These studies reported different results from our study, in which IgG, IL-10, and TNF-α were not affected by the experimental diet.

The hematological and biochemical parameters of the dogs were analyzed to confirm the safety of BSFL and FO. All the hematological parameters were within the normal reference ranges in all the experimental groups. In this study, WBC at the end of the experiment was significantly higher in the BSFL group than in the CON group, and the level of BASO in the BSFL group was significantly increased over time (*p* < 0.05). WBC and BASO are two of the indicators monitored to indicate changes in pathological conditions accompanied by hypersensitivity, and allergic and inflammatory reactions [51]. Although our results showed that the WBC and BASO levels in the BSFL group were significantly different (vs. CON) and changed over time, this does not mean that BSFL caused pathological problems, because these levels still remained within the normal range throughout the experimental period. Furthermore, this claim is supported by the results for the BW, skin status, fecal score, and immune-related parameters mentioned above. Additionally, these results are consistent with those of Kröger et al. and Freel et al., who reported that feeding dogs BSFL did not negatively affect their hematological parameters [17,18]. To the best of our knowledge, studies on using fermented oats as raw materials for dog food appear limited. Gizzarelli et al. reported that dogs fed an oat-based diet showed no significant changes in hematological parameters (WBC, RBC, HGB, HCT, MCV, and MCH) compared to dogs fed a rice-based diet [52]. Although their study focused on oats, their results are consistent with those of our study using FO.

Some biochemical parameters (PHOS, GLOB, and ALKP) showed significant changes among the CON and each treatment group at the end of the trial when feeding with FO, BSFL, and their combination. However, most parameters were observed to be within the normal reference ranges, with GLOB and T-BIL being the exceptions. Although the concentrations of serum GLOB and T-BIL were outside the normal reference ranges, the reference range presented in this study is a simple reference value provided by the analysis equipment (Hitachi 7180; Hitachi High-Technologies Co., Tokyo, Japan), and the reference ranges presented in this paper are not absolute criteria for judging whether a dog is clinically and pathologically normal or abnormal. In addition, different standards for the reference range of serum biochemistry in animals are suggested for different research institutions, analysis equipment, and researchers, and it has been argued that different optimal normal ranges should be applied depending on the breed, age, and physiological state, among other factors [53,54,55]. When the reference ranges suggested by Fielder [53] and Dall’Aglio [54] were applied, the concentrations of serum GLOB and T-BIL in this study were within the normal ranges. Furthermore, the experimental dogs in this study were judged to have a normal health condition, under the diagnosis of a professional veterinarian, based on the results of their hematologic and biochemical parameters and clinical health status at the time of trial initiation. Whalan suggested that it would be ideal to consider correlations with other parameters rather than just one anomalous parameter when utilizing clinical pathology data to evaluate animal health status [51]. Other parameters associated with GLOB and T-BIL (T-PRO and albumin in the case of GLOB and CBC in the case of T-BIL) were within the normal range in this study. In addition, the pathological signs associated with their parameters (acute inflammatory disease and atopic dermatitis, etc., for GLOB, and inflammation, shock, and excessive hemolysis for T-BIL) were not observed. By comprehensively considering all the factors mentioned, we judged that all dogs used in this study had normal physiological conditions.

Several studies have reported the results of evaluating the effects of BSFL and oat feeding on serum biochemical parameters. Freel et al. reported that the concentration of ALT in the serum of adult beagles was significantly increased by BSFL feeding for 28 days [18], and Lei et al. reported a linear increase in the ALB concentration depending on the amount of BSFL (0%, 1%, or 2%) [15]. Gizzarelli et al. reported that biochemical parameters (GLU, CREA, BUN, PHOS, CA, T-PRO, ALB, GLOB, ALT, GGT, and CHOL) were not significantly affected by feeding a diet containing oats in healthy adult dogs [52]. The inconsistency among the results of those studies and this study is considered to be caused by various factors, such as the concentration of materials, the feeding period, the animal age, the animal breed, etc. Thus, further studies are needed to determine the effects of BSFL and FO on serum biochemical characteristics in dogs.

Notably, the BSFL group showed a significant decrease in T-CHO over the 12 weeks, while the other groups showed slight increases when comparing the values at the start and end of the experiment, though these were not significant. One of the nutritional properties of BSFL is that, similar to coconut oil, it has a high content of lauric acid, which is a medium-chain fatty acid (C12). The BSFL used in this study contained about 26.9% fat (Table 1), and the lauric acid content among the total fatty acids was 32.8 g/100 g (data not shown). Wood and Migicovsky reported that lauric acid reduces the incorporation of cholesterol in the liver, whereas unsaturated oils increased the total cholesterol in rat liver [56]. In addition, in a human feeding study, Hashim et al. observed that medium-chain triglyceride prepared with C6–C12 saturated fatty acid temporarily elevated and then reduced the serum cholesterol [57]. The finding of our study that serum cholesterol was reduced in dogs fed a diet supplemented with BSFL is consistent with their results. However, other studies reported conflicting results, whereas all of the saturated fatty acids (C8 to C16), including lauric acid, increased total cholesterol [58,59]. The effect of lauric acid on serum cholesterol is still controversial because it appears to be influenced by various factors, such as the duration of the experiment [60]. To resolve this, further studies on whether BSFL can reduce serum total cholesterol in dogs with hypercholesterolemia are required.

## 5. Conclusions

This study was conducted to evaluate the suitability of FO and BSLF as food materials for dogs. Comprehensively, the feeding of 10% FO and 5% BSFL for 12 weeks did not affect food intake, body weight, or BCS, and did not have a negative effect on physiological and biochemical responses in dogs. Furthermore, the findings suggest that BSFL may have the ability to reduce serum total cholesterol in dogs. Further studies of the effects of BSFL on serum total cholesterol in dogs are required. Our results demonstrate the safety and potential functionality of FO and BSFL, and verify their suitability as food ingredients for dogs.

## Figures and Tables

**Figure 1 animals-11-03509-f001:**
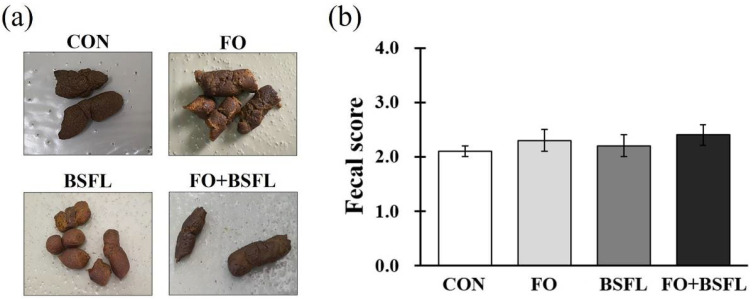
Effects of dog food with FO and BSFL on fecal scores of dogs: (**a**) photographs of the feces; (**b**) scores based on the following 5-point fecal score scale (1 = hard and dry feces to 5 = liquid diarrhea). The results are expressed as mean ± SE. CON, control group; FO, group with 10% fermented oat added to food; BSFL, group with 5% black soldier fly larva added to food; FO + BSFL, group with 10% fermented oat and 5% black soldier fly larva added to food. The *p*-value on fecal score was 0.666.

**Figure 2 animals-11-03509-f002:**
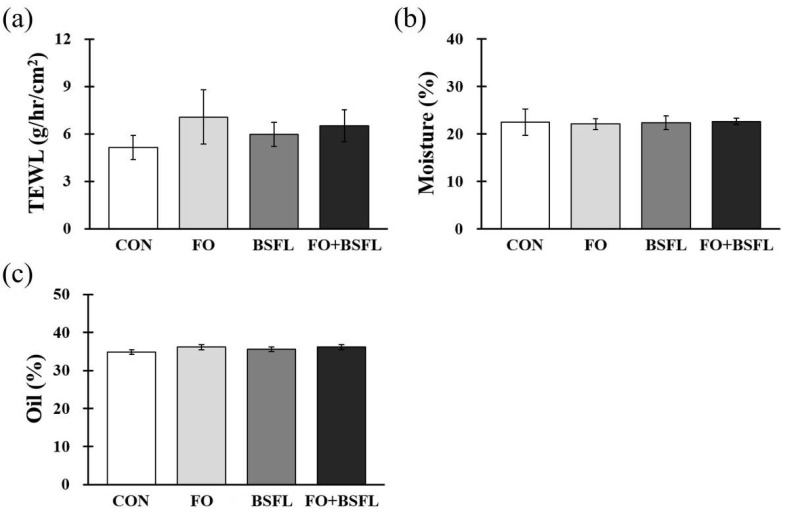
Effects of dog food with FO and BSFL on skin barrier status in dogs. (**a**) TEWL, (**b**) moisture, and (**c**) oil concentration were measured from four different areas (groin skin, armpit skin, back skin, and ear skin) at the end of the experiment. The results are expressed as mean ± SE. TEWL, transepidermal water loss. CON, control group; FO, group with 10% fermented oat added to food; BSFL, group with 5% black soldier fly larva added to food; FO + BSFL, group with 10% fermented oat and 5% black soldier fly larva added to food. *p*-value: (**a**) 0.486, (**b**) 0.565, (**c**) 0.639.

**Figure 3 animals-11-03509-f003:**
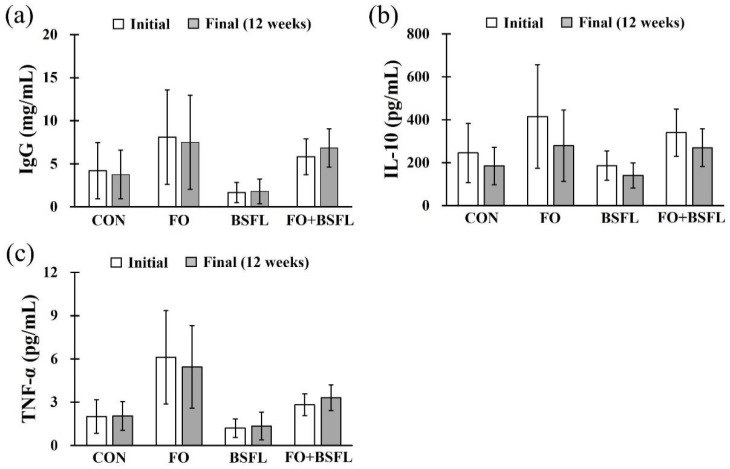
Effect of dog food with FO and BSFL on the IgG and inflammatory cytokines in dogs: (**a**) IgG, (**b**) IL-10, and (**c**) TNF-α. The results are expressed as mean ± SE. IgG, immunoglobulin G; IL-10, interleukin 10; TNF-α, tumor necrosis factor alpha. CON, control group; FO, group with 10% fermented oat added to food; BSFL, group with 5% black soldier fly larva added to food; FO + BSFL, group with 10% fermented oat and 5% black soldier fly larva added to food.

**Table 1 animals-11-03509-t001:** Analyzed chemical composition of experimental foods.

Items	Ingredients		Groups ^1^			
	FO	BSFL	CON	FO	BSFL	FO + BSFL
Ingredients (%)						
Rice powder			58.03	48.82	56.07	47.23
Poultry meal			22.5	22.0	20.0	19.22
BSFL			-	-	5	5
FO			-	10	-	10
Egg yolk powder			13.12	13.08	13.13	13.25
Calcium carbonate			1.35	1.3	1.0	0.9
Lard			1.0	1.0	0.9	0.6
Cabbage powder			1.0	1.0	1.0	1.0
Seaweed (*Enteromorpha*) powder			1.0	1.0	1.0	1.0
Potassium citrate			0.6	0.6	0.6	0.6
Vit. and Min. premix ^2^			0.5	0.5	0.5	0.5
Salt			0.4	0.4	0.4	0.4
Calcium phosphate			0.3	0.3	0.4	0.3
Analyzed composition (%DM)						
Moisture	4.36	3.62	10.9	10.24	10.18	12.21
Crude protein (CP)	18.95	46.83	31.32	32.11	31.17	31.79
Ether extract (EE)	8.82	26.87	11.65	12.19	12.51	13.01
Crude fiber (CF)	3.69	12.71	0.19	0.78	0.86	1.44
Crude ash (CA)	1.62	10.68	2.27	2.36	2.73	2.83
Nitrogen-free extract (NFE)	66.92	2.91	54.58	52.55	52.72	50.93
Calcium (Ca)	0.08	3.6	0.68	0.66	0.75	0.69
Phosphorus (P)	0.38	0.67	0.46	0.44	0.50	0.46
Ca/P ratio	0.21	5.37	1.47	1.47	1.50	1.49
Calculated ME, kcal/kg ^3^	3755	4025	3997	3999	4000	4001

^1^ CON, control group; FO, group with the addition of 10% fermented oat; BSFL, group fed with 5% black soldier fly larva added; FO + BSFL, group fed with the addition of 5% black soldier fly larva and 10% fermented oat. ^2^ Vitamin and mineral premix was supplied per kilogram of diets at 3500 IU of vitamin A; 250 IU of vitamin D3; 25 mg of vitamin E; 0.052 mg of vitamin K; 2.8 mg of vitamin B1 (thiamine); 2.6 mg of vitamin B2 (riboflavin); 2 mg of vitamin B6 (pyridoxine); 0.014 mg of vitamin B12; 6 mg of Cal-d-pantothenate; 30 mg of niacin; 0.4 mg of folic acid; 0.036 mg of biotin; 1000 mg of taurine; 44 mg of FeSO_4_; 3.8 mg of MnSO_4_; 50 mg of ZnSO_4_; 7.5 mg of CuSO_4_; 0.18 mg of Na_2_SeO_3_; 0.9 mg of Ca(IO_3_)_2_. ^3^ ME, metabolizable energy (kcal/kg) = ((CP × 3.5) + (EE × 8.5) + (NFE × 3.5)) × 10.

**Table 2 animals-11-03509-t002:** Effects of dog food with FO and BSFL on food intake and body parameters in dogs at the beginning (initial) and end (final) of the experiment.

Items	CON ^1^	FO	BSFL	FO + BSFL	*p*-Value
ADFI ^2^, g/d	98.0 ± 9.2	100.4 ± 12.0	102.3 ± 10.2	100.2 ± 12.0	0.735
Body weight, kg					
Initial	4.13 ± 0.75	4.22 ± 0.67	4.20 ± 0.67	4.17 ± 0.67	0.999
Final	4.49 ± 0.83	4.52 ± 0.76	4.62 ± 0.76	4.51 ± 0.73	0.999
Rate of BWG ^3^, %	108.7 ± 1.2	106.5 ± 1.2	109.8 ± 2.1	108.0 ± 1.0	0.437
BCS ^4^					
Initial	4.20 ± 0.73	4.20 ± 0.58	3.60 ± 0.93	3.40 ± 0.87	0.850
Final	4.60 ± 0.68	4.20 ± 0.49	3.60 ± 0.81	3.40 ± 0.68	0.593

Values are expressed as mean ± SE. ^1^ CON, control group; FO, group with 10% fermented oat added to food; BSFL, group with 5% black soldier fly larva added to food; FO + BSFL, group with 10% fermented oat and 5% black soldier fly larva added to food. ^2^ ADFI, average daily food intake; ^3^ BWG, body weight gain; ^4^ BCS, body condition score.

**Table 3 animals-11-03509-t003:** Effects of dog food with FO and BSFL on CBCs in dogs.

Items	CON	FO	BSFL	FO + BSFL	*p*-Value
WBC, ×10^6^/mL (Ref. range: 5.05–16.76)					
Initial	8.08 ± 0.80	8.10 ± 1.89	10.26 ± 2.15	6.28 ± 1.00	0.385
Final	7.48 ± 0.81	9.48 ± 2.88	12.24 ± 1.10 *	7.65 ± 1.08	0.194
NEU, ×10^6^/mL (Ref. range: 2.95–11.64)					
Initial	5.42 ± 0.53	4.97 ± 1.23	7.23 ± 1.71	4.11 ± 0.86	0.321
Final	5.58 ± 0.54	6.56 ± 2.21	8.38 ± 1.16	5.57 ± 0.64	0.415
LYM, ×10^6^/mL (Ref. range: 1.05–5.10)					
Initial	1.69 ± 0.28	2.15 ± 0.38	1.82 ± 0.29	1.40 ± 0.16	0.362
Final	1.48 ± 0.26	2.06 ± 0.39	2.74 ± 0.79	1.34 ± 0.30	0.200
MONO, ×10^6^/mL (Ref. range: 0.16–1.12)					
Initial	0.55 ± 0.24	0.65 ± 0.25	0.65 ± 0.25	0.46 ± 0.13	0.919
Final	0.19 ± 0.12	0.55 ± 0.29	0.74 ± 0.20	0.53 ± 0.18	0.324
EOS, ×10^6^/mL (Ref. range: 0.06–1.23)					
Initial	0.43 ± 0.07	0.33 ± 0.10	0.56 ± 0.22	0.31 ± 0.08	0.542
Final	0.06 ± 0.01	0.16 ± 0.05	0.27 ± 0.15	0.14 ± 0.06	0.406
BASO, ×10^6^/mL (Ref. range: 0–0.1)					
Initial	0.00 ± 0.00	0.002 ± 0.00	0.004 ± 0.00	0.00 ± 0.00	0.547
Final	0.05 ± 0.02	0.11 ± 0.05	0.10 ± 0.03 ^#^	0.07 ± 0.03	0.516
RBC, ×10^9^/mL (Ref. range: 5.65–8.87)					
Initial	5.85 ± 0.26	5.84 ± 0.17	6.11 ± 0.09	5.88 ± 0.27	0.783
Final	6.27 ± 0.31	6.01 ± 0.11	6.33 ± 0.18	6.02 ± 0.31	0.702
HGB, g/dL (Ref. range: 13.1–20.5)					
Initial	14.26 ± 0.61	13.94 ± 0.47	14.32 ± 0.44	14.16 ± 0.64	0.963
Final	14.40 ± 0.69	13.82 ± 0.54	14.14 ± 0.41	14.18 ± 0.99	0.947
HCT, % (Ref. range: 37.3–61.7)					
Initial	41.28 ± 1.65	40.36 ± 1.03	42.24 ± 1.06	40.52 ± 1.76	0.777
Final	44.51 ± 1.98	42.27 ± 0.88	43.89 ± 0.93	43.16 ± 2.25	0.790

Values are expressed as mean ± SE. WBC, white blood cell; NEU, neutrophils; LYM, lymphocytes; MONO, monocytes; EOS, eosinophils; BASO, basophils; RBC, red blood cells; HGB, hemoglobin; HCT, hematocrit. *, significant differences from the control in the same row (*p* < 0.05); ^#^, significant differences between the initial and final values in the same column (*p* < 0.05). CON, control group; FO, group with 10% fermented oat added to food; BSFL, group with 5% black soldier fly larva added to food; FO + BSFL, group with 10% fermented oat and 5% black soldier fly larva added to food.

**Table 4 animals-11-03509-t004:** Effects of dog food with FO and BSFL on serum biochemical parameters in dogs.

Items	CON	FO	BSFL	FO + BSFL	*p*-Value
GLU, mg/dL (Ref. range: 70–138)					
Initial	97.2 ± 8	98.6 ± 4.34	103.6 ± 6.45	95.8 ± 6.07	0.835
Final	94.6 ± 4.58	95.6 ± 4.34	91.8 ± 3.12	97 ± 3.51	0.816
CREA, mg/dL (Ref. range: 0.5–1.6)					
Initial	0.64 ± 0.09	0.64 ± 0.07	0.68 ± 0.05	0.6 ± 0.05	0.883
Final	0.72 ± 0.12	0.72 ± 0.09	0.72 ± 0.07	0.68 ± 0.07	0.984
BUN, mg/dL (Ref. range: 6.0–31)					
Initial	13.6 ± 0.93	14.4 ± 1.63	16.4 ± 1.47	15 ± 1.38	0.545
Final	14.6 ± 1.25	15 ± 1.22	15.8 ± 1.2	14 ± 1.41	0.789
PHOS, mg/dL (Ref. range: 2.5–6.0)					
Initial	4.74 ± 0.35	4.62 ± 0.37	4.22 ± 0.29	4.32 ± 0.32	0.655
Final	4.54 ± 0.26	4.54 ± 0.47	3.52 ± 0.31 *	3.72 ± 0.25	0.049
CA, mg/dL (Ref. range: 8.9–11.4)					
Initial	9.18 ± 0.41	9.34 ± 0.29	9.64 ± 0.48	9.64 ± 0.21	0.295
Final	9.08 ± 0.36	9.38 ± 0.16	8.96 ± 0.32	8.9 ± 0.23	0.628
T-Pro, g/dL (Ref. range: 5.0–7.4)					
Initial	6.98 ± 0.37	6.76 ± 0.27	7.24 ± 0.17	6.24 ± 0.2	0.089
Final	7.34 ± 0.26	6.66 ± 0.25	6.62 ± 0.19 ^#^	6.56 ± 0.22	0.100
ALB, g/dL (Ref. range: 2.7–4.4)					
Initial	3.16 ± 0.12	3.22 ± 0.08	3.44 ± 0.11	2.94 ± 0.14	0.087
Final	3.22 ± 0.09	3.2 ± 0.09	3.14 ± 0.12	3.14 ± 0.13	0.935
GLOB, g/dL (Ref. range: 1.6–3.6)					
Initial	3.82 ± 0.28	3.54 ± 0.22	3.8 ± 0.17	3.3 ± 0.13	0.271
Final	4.12 ± 0.23	3.46 ± 0.29	3.48 ± 0.19	3.42 ± 0.12 *	0.048
A/G ratio (Ref. range: 0.8–2.0)					
Initial	0.84 ± 0.05	0.92 ± 0.05	0.92 ± 0.07	0.88 ± 0.06	0.715
Final	0.78 ± 0.06	0.98 ± 0.09	0.92 ± 0.07	0.9 ± 0.03	0.221
ALT, U/L (Ref. range: 12–118)					
Initial	111.4 ± 48.17	58.8 ± 13.75	135.5 ± 43.56	41.2 ± 6.18	0.164
Final	87 ± 31.58	38.4 ± 9.9	97.5 ± 35.26	35.4 ± 8.07	0.246
ALKP, U/L (Ref. range: 5.0–131)					
Initial	60.6 ± 19.94	24.6 ± 3.61	62 ± 17.52	20.6 ± 4.55	0.079
Final	58.4 ± 7.93	36.4 ± 4.17 *	70 ± 24.03	29 ± 5.39 *	0.043
GGT, U/L (Ref. range: 0–12)					
Initial	0 ± 0	0 ± 0	3 ± 3	0 ± 0	0.418
Final	0 ± 0	0 ± 0	2.8 ± 2.8	0 ± 0	0.418
T-BIL, mg/dL (Ref. range: 0.1–0.3)					
Initial	0.28 ± 0.04	0.22 ± 0.04	0.4 ± 0.03 *	0.3 ± 0.03	0.004
Final	0.28 ± 0.04	0.2 ± 0.04	0.3 ± 0.05	0.2 ± 0.03	0.254
CHOL, mg/dL (Ref. range: 29–291)					
Initial	131.4 ± 15.16	136.4 ± 8.89	177.6 ± 8.95 *	132.8 ± 6.58	0.017
Final	143.2 ± 18.99	141.8 ± 10.94	159.4 ± 9.89	153 ± 10.29	0.749

Values are expressed as mean ± SE. GLU, glucose; CREA, creatinine; BUN, blood urea nitrogen; PHOS, phosphorous; CA, calcium; T-Pro, total protein; ALB, albumin; GLOB, globulin; A/G, albumin/globulin ratio; ALT, alanine aminotransferase; ALKP, alkaline phosphatase; GGT, gamma glutamyltransferase; T-BIL, total bilirubin; T-CHO, total cholesterol. *, significant difference from the control in the same row (*p* < 0.05); ^#^, significant differences between initial and final values in the same column (*p* < 0.05). CON, control group; FO, group with 10% fermented oat added to food; BSFL, group with 5% black soldier fly larva added to food; FO + BSFL, group with 10% fermented oat and 5% black soldier fly larva added to food.

## Data Availability

Not applicable.

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
