# Peer review of "Evaluation of Fermented Oat and Black Soldier Fly Larva as Food Ingredients in Senior Dog Diets"

_animals, 2021, doi:10.3390/ani11123509_

Round 1

Reviewer 1 Report

I would like to thank the authors for allowing me to review this manuscript revision. I feel that the authors have adequately addressed deficiencies from the first round of edits, and I congratulate them on an acceptable manuscript.

Author Response

Dear Reviewer 1

Thank you for your review and comments.

I am glad that your review could improve the quality of this manuscript.

Kind regards,

All authors

Reviewer 2 Report

Dear Authors

I attached the review, suggestion and question with this document.

Please, carefully replied and corrected. Any question, please feel free to ask.

Author Response

Dear Reviewer 2

Thank you for your review and valuable comments.

I am glad that your comments could improve the quality of this manuscript.

This manuscript was revised according to your comments as follows. (in red)

The typo was corrected. (L.38)
Added some words suggested by reviewer. (L.3, 90-91, 97-98, 106, 175, 200-201, 208, and Table 1)
Title has been corrected. (L.3)
The p-values was represented in all tables and figures in the results section.

However, the statistical analysis was performed by suggested way, we confirmed that the results still be the same as present manuscript. Therefore, we did not change on statistical analysis.

Thank you

Kind regards,

All authors

This manuscript is a resubmission of an earlier submission. The following is a list of the peer review reports and author responses from that submission.

Round 1

Reviewer 1 Report

As there are no line numbers present, it is difficult to refer to where changes should be made. The authors should pay close attention to formatting details and make advantageous use of the provided manuscript template.

2.2 How were the inclusion rates of FO and BSFL determined? Was their a basis for these inclusion rate? Previous literature? My suspicion is that these values were chosen such that rations would be both isonitrogenous and isocaloric (based on Table 1), but this should be clearly stated.

2.4 Given that data were analyzed in SPSS, how could an ANCOVA be calculated in PROC GLM, a SAS procedure? Also, how could breed be used as a covariate given than an ANCOVA implies the use of a continuous covariate effect?

3.1 It appears that BCS was analyzed as a continuous variable. Is this appropriate, given that BCS is an ordinal value for which there are no continuous intermediate values? The same can be said about the fecal scores on a scale of 1-5.

4.1 The authors make comparisons between FO and oats, which I find very helpful. However, the statement "Fermented feed is known to be less palatable to dogs due to its unique smell and taste" is a bit concerning. I agree with this conclusion, and the statement is well cited. However, since the FO were obtained through centrifugation, wouldn't some of those offensive compounds be removed in the supernatant? I feel that this inquiry would add to the discussion.

Author Response

Dear Reviewer 1

We thank  for your comments, which have helped us to improve our manuscript.

We are now submitting our responses to the your comments and revised manuscript with attached file.

Please find an uploaded file.

Kind regards,

Kihyun Kim

Manuscript ID: animals-1426088

Title: Evaluation of Fermented Oat and Black Soldier Fly Larva as Feed Ingredients in Dog Diets

Authors: Kangmin Seo, Hyun-Woo Cho, Julan Chun, Junghwan Jeon, Chanho Kim, Kwanho Park, Kihyun Kim

Reviewer 2 Report

General comments

You forgot to number the lines before you submitted the manuscript. Please add the lines.

Please consider replacing feed by food.

Keywords: I would encourage the authors to think on keywords that do not appear in the title of the manuscript. This would increase the chances of your manuscript to be found in a search engine.

Abstract: please describe all the abbreviation used in the abstract. This is something you must do throughout the manuscript. Please revise accordingly

Specific comments

Last paragraph Intro – please remove “grade” after “organic”

Last paragraph Intro – please add the species that were used in the trials from references 20-24. If these trials were not done in dogs, you cannot extend the results to dogs.

Objective statement – you need to make clear that you are measuring the effects of FO and BSFL in dogs, not dog diets. This is not a pet food processing work, this is nutrition. Please revise accordingly.

2.1. BSFL and FO Prep – you must declare the concentration of each bacterium that was added to the oats.

2.2. Animals, Designs, Diets, and Housing – why did you use only female dogs? This must be further explained in this session.

2.2. Animals, Designs, Diets, and Housing – how were these diets made? You need to provide more details on how the diets were made. How were the ingredients mixed? For how long? How was the food made? Extrusion? Canning? Was the food dried? Was any fat added after the food was dried? Was a palatant added? Please revise.

2.2. Animals, Designs, Diets, and Housing – please cite all the methods used for all nutrient analysis reported on Table 1.

Table 1 – was NFE calculated or analyzed? Please explain

Table 1 – Your ME equation is wrong. For example, for you CON diet 31.32*3.5 + 11.65*8.5 + 54.58*3.5 = 399.675 and not 3997 as reported. Please revise.

2.2. Animals, Designs, Diets, and Housing – was food intake adjusted to maintain body weight or the amount estimated by the MER equation was fed throughout the duration of the trial?

2.2. Animals, Designs, Diets, and Housing – the Laflamme (1997) citation is referenced as stated by Animals guidelines. Please revise and also check all your references for compliance with the journal’s guidelines.

2.3. Sampling and Analysis – please combine all the information related to CBC together and then all the information regarding serum chemistry together.

2.3. Sampling and Analysis – please provide detailed information on how the TEWL, moisture, and oil content were measured. Was the analyzed skin site shaved? Was the site cleaned somehow?

3.1. Feed Intake, Body Parameters, and Fecal Score – please use among treatments instead of between. You should only use between then you are referring to 2 things. For example, between CON and FO. Otherwise, you should always use among. For example, no significant differences were found among the CON and each treatment group.

Table 2. I don’t think the diets were supplemented with FO or BSFL, I think the appropriated title for this table would be “Effect of dog food with fermented oat and …”

Table 2 Please add superscripts on this table to address all the abbreviations in the footnotes. You must describe how rate of BWG was calculated in the materials and method. Please revise accordingly.

Remove the sentence “The fecal scores were evaluated with a 5-point fecal score scale (1= hard and dry feces to 5 = liquid diarrhea).” This is part of Materials and Methods and not Results of your research.

3.3. Hematological and Biochemical Parameters – You should start this sentence with data is presented on tables 3 and 4, everything was withing the reference range, except… First you need to report what was not within the reference range, then you can dive into the differences among treatments or collection points. Since most of your analyzed parameters were within the normal range, the differences among treatments are not as relevant as the collections points that these parameters were outside the reference range. Because the values were out the reference range, the animals should not be considered healthy and should have been either removed from the study or a detailed explanation for their continuation on the trial must be provided.

4.1. Feeding and Body Parameters – in addition to comparing FO with oats, you should also consider comparing FO with other fermented ingredients, or ingredients that are generated from the fermentation industry. Or ingredients that would have a composition somewhat similar to FO.

4.2. Safety and Health Parameters - you never mentioned allergy related to protein intake anywhere in your introduction. While you somewhat describe how the allergic response happens in dogs, you never explain that protein allergy is something that dogs can develop through exposure to the protein source. The most common protein allergy in dogs if chicken, and the most used protein source in dog food is chicken-based.

4.2. Safety and Health Parameters - You need to organize much better your data discussion here. You should follow an order that is somewhat similar to the order you presented the data in the Results section. You should also look into other papers that evaluated some sort of fermented ingredient at similar inclusion levels. Once you ferment the oats, likely there are several fermentation products that are formed and may influence some of the results you reported. By referencing papers that used fermented products, you would have a better basis to compare results rather than just a paper that used oats.

4.2. Safety and Health Parameters – why did you consider these values for GLOB and T-BIL normal? What are your references for that? You MUST explain this in a lot more detail. How do you know that it is a dog variation rather than an effect of the food or if the dog has some underlying condition? You need a lot more here to be able to report these values as normal.

4.2. Safety and Health Parameters – you need a better explanation for the differences in parameters among your and other published literature. It is expected that you would report a more differences because the inclusion level was higher in your study compared to previous reports. Please, look into possible causes for these differences rather than the experimental conditions.

Conclusions – you cannot conclude anything about “digestive physiology” of the inclusion of FO or BSFL in dogs, because you did not measure any digestion related parameter in this trial. I would caution you to have strong statements such as “Our study findings confirm that long-term (12 weeks) feeding of 10% FO and 5% BSFL did not induce food allergy” You only fed 15 dogs this food. And Protein-related food allergies are not likely to occur in healthy dogs. This sentence should be removed.

Author Response

Dear Reviewer 2

We thank for your comments, which have helped us to improve our manuscript.

We are now submitting our responses to the your comments and revised manuscript with attached file.

Please find an uploaded file.

Kind regards,

Kihyun Kim
